behaviour, developmental biology, genetics

nitrogen narcosis, hyperbaric chamber, dopamine, *cat-2*, *dat-1*, *C. elegans*

**Author for correspondence:**
Limor Broday
e-mail: broday@tauex.tau.ac.il

# Dopamine-dependent biphasic behaviour under 'deep diving' conditions in *Caenorhabditis elegans*

Inbar Kirshenboim[1,2], Ben Aviner[2], Eyal Itskovits[3], Alon Zaslaver[3] and Limor Broday[1]

[1]Department of Cell and Developmental Biology, School of Medicine, Tel Aviv University, Tel Aviv, Israel
[2]Israel Naval Medical Institute, Israel Defense Forces Medical Corps, Haifa, Israel
[3]Department of Genetics, Silberman Institute of Life Science, Edmond J. Safra Campus, The Hebrew University of Jerusalem, Jerusalem 9190401, Israel

LB, 0000-0002-3864-0500

Underwater divers are susceptible to neurological risks due to their exposure to increased pressure. Absorption of elevated partial pressure of inert gases such as helium and nitrogen may lead to nitrogen narcosis. Although the symptoms of nitrogen narcosis are known, the molecular mechanisms underlying these symptoms have not been elucidated. Here, we examined the behaviour of the soil nematode *Caenorhabditis elegans* under scuba diving conditions. We analysed wild-type animals and mutants in the dopamine pathway under hyperbaric conditions, using several gas compositions and under varying pressure levels. We found that the animals changed their speed on a flat bacterial surface in response to pressure in a biphasic mode that depended on dopamine. Dopamine-deficient *cat-2* mutant animals did not exhibit a biphasic response in high pressure, while the extracellular accumulation of dopamine in *dat-1* mutant animals mildly influenced this response. Our data demonstrate that in *C. elegans*, similarly to mammalian systems, dopamine signalling is involved in the response to high pressure. This study establishes *C. elegans* as a powerful system to elucidate the molecular mechanisms that underly nitrogen toxicity in response to high pressure.

## 1. Introduction

When humans are exposed to compressed air or a nitrogen–oxygen mixture at a pressure of 3–4 atmospheres (0.3–0.4 MPa) or in a depth of 10–30 m during diving, they may experience nitrogen narcosis which is a life-threatening condition. Nitrogen narcosis is defined as a reversible change in neuromuscular function affecting consciousness and behaviour that occur during diving at deep depths while breathing compressed inert gases [1]. The common symptoms of nitrogen narcosis include disorientation, euphoria, anxiety, hallucinations, impaired coordination and more. This can lead to loss of consciousness and death [2]. The symptoms differ between divers and disappear as soon as the diver ascends from the depth at which the symptoms became apparent. The underlying mechanisms for nitrogen narcosis remain unclear although this phenomenon is common as it occurs at recreational depth and could place the diver in a life-threatening situation. Several studies suggest that the site of action of hyperbaric nitrogen are hydrophobic sites on various membrane proteins mainly of ligand-gated ion channels (GABAA receptors, glutamate receptors), voltage-gated ion channels, G-protein coupled receptors and other ion channels expressed in the central nervous system, causing disruptions of inhibitory and excitatory synaptic functions [3,4].

As dopamine plays a critical role in animal's response to changing environmental stimuli [5], it has been studied in relation to nitrogen narcosis [6]. Indeed, studies in rats have shown that hyperbaric nitrogen decreases the release

of neurotransmitters such as glutamate and dopamine, impairs the nigrostriatal pathway and leads to desensitization of GABA receptors [4,7]. In addition, decrease in dopamine levels in rats is suppressed by recurrent exposure to nitrogen narcosis conditions, suggesting an adaptation mechanism [4,8]. This adaptive phenomenon has been examined recently in expert scuba divers at 0.6 MPa [9]. Dopamine is a major neuromodulator in multicellular organisms and dopamine signalling is highly conserved along evolution [10]. The simple anatomy and the genetics of the nematode *Caenorhabditis elegans* has made it an excellent model for studying behavioural plasticity [11]. The adult *C. elegans* hermaphrodite has 302 neurons for which the connectivity was fully mapped [12,13]. The hermaphrodite has eight dopaminergic neurons: two ADE neurons (anterior sensory neurons), four CEP neurons (anterior sensory neurons) and two PDE neurons (posterior sensory neurons) [14]. Additional six dopaminergic neurons are located in the tail of the male [14]. Laser ablation of the dopaminergic neurons or mutations that prevent synthesis or release of dopamine cause defects in the ability of the animals to sense and respond to environmental changes [15]. Dopaminergic signalling in *C. elegans* controls many behaviours including locomotion [16–18], egg laying [19,20] and associative learning [21]. *Caenorhabditis elegans* is typically fed on a bacterial lawn and its speed can be modulated by the bacteria. Wild-type (WT) animals move relatively slowly in the presence of bacteria when compared to their velocity in the absence of bacteria. This basal slowing response is mediated by dopamine. In *C. elegans*, dopamine is synthesized by the tyrosine hydroxylase enzyme CAT-2 [22], which catalyses the conversion of tyrosine to L-DOPA, the biosynthetic precursor of dopamine [23]. Dopamine is sequestered in presynaptic storage vesicles by the vesicular monoamine transporter CAT-1. It is stored in these vesicles until released following neuronal depolarization. In the synapse, dopamine binds and activates D1-like (DOP-1) and D-2 like receptors (DOP-2 and DOP-3). Unbound dopamine is transported back into the presynaptic cell by the dopamine re-uptake transporter DAT-1. Viable dopamine-deficient mutants make *C. elegans* a multicellular model suitable for studying the molecular and the physiological effects of impaired dopamine signalling. In this study, we focused on two mutants in the dopamine pathway: *cat-2* and *dat-1*. CAT-2 is expressed in all dopaminergic neurons of the nematode [22]. The *cat-2* mutant animals have reduced levels of dopamine (9% compared to WT) [23] and these low dopamine levels are probably maintained by the activity of tyrosinase enzymes that are expressed in the dopaminergic neurons [15]. The *cat-2* mutants move faster than WT animals on a bacterial lawn [15–18]. DAT-1 is a dopamine transporter that regulates dopamine levels by re-uptake of free dopamine. DAT-1 has, therefore, an important role in negative regulation of dopamine signalling through its clearance [24]. Excess dopamine signalling in *dat-1* mutants reduces crawling speed on solid media and induces a swimming paralysis (Swip) in liquid media [25–27]. We compared the response of WT animals to *cat-2* and *dat-1* mutant worms under varying pressures and three different gas compositions (table 1). To analyse the effect of pressure on animal's velocity, we used a video-tracking system that allowed extracting trajectories from multiple freely behaving animals [28]. Our study revealed that in *C. elegans*, dopamine controls locomotion speed in response to high pressure. Both compressed air and

**Table 1.** Summary of the gas compositions and the partial pressures tested in this study.

| gas | pressure (Mpa) | partial pressure of nitrogen (Mpa) | partial pressure of oxygen (Mpa) |
| --- | --- | --- | --- |
| air | 0.220 | 0.174 | 0.044 |
| | 0.700 | 0.553 | 0.140 |
| | 1.100 | 0.869 | 0.220 |
| oxygen | 0.100 | 0.000 | 0.100 |
| | 0.220 | 0.000 | 0.220 |
| nitrogen | 0.100 | 0.095 | 0.050 |
| mixture | 1.100 | 1.045 | 0.055 |

compressed nitrogen induced changes in animal speed while compressed oxygen did not affect locomotion speed in both WT and dopamine mutant animals.

## 2. Results

### (a) Biphasic speed response to increased air pressure in wild-type *Caenorhabditis elegans*

Worms were exposed to pressure in a hyperbaric chamber (figure 1*a*–*c*) and imaged in 1 s intervals. The average locomotion speed was quantified using multi-animal-tracker (MAT) [28]. The MAT tool determined the positions of multiple animals simultaneously followed by data assembly into tracks (figure 1*d* ). We first analysed the changes in the average speed of WT animals during compression to 1.1, 0.7 and 0.22 MPa in air. Imaging of the animals inside the chamber was performed in accordance with a cycle of a 'diving' protocol: before compression, during compression (0.03 MPa min$^{-1}$), at stable high pressure, decompression (0.03 MPa min$^{-1}$) and recovery. When the pressure in the chamber reached the tested value, the worms were kept inside the chamber under this pressure for 1 h. The average speed was quantified using the MAT system and was normalized to the average control speed in normobaric conditions (0.1 MPa air). When testing a maximal pressure of 1.1 and 0.7 MPa, we observed a biphasic behaviour (figure 2*a* and electronic supplementary material, table S1). The biphasic behaviour [29] is a behaviour of hyperactivity (high locomotor and motor activity), followed by a decrease of activity. We detected both phases, while the second phase of decreased activity was observed during the exposure to high pressure, or even during compression. In the first phase during compression, we observed an increase in animal's velocity. In 1.1 MPa, the maximum normalized speed relative to control was $4.9 \pm 0.5$ ($p < 0.0001$) and was detected at 0.9 MPa before reaching 1.1 MPa inside the chamber (figure 2*a*, orange line and electronic supplementary material, movie S1). In the condition of 0.7 MPa the maximum normalized speed value was $2.7 \pm 0.5$ ($p < 0.0001$) measured at 0.7 MPa (figure 2*a*, blue line). Interestingly, when comparing the second phase of the biphasic response, the speed of the worms decreased during exposure to a pressure of 1.1 MPa but remained relatively unchanged under the constant pressure of 0.7 MPa. These data show that the level of the biphasic phase depends on

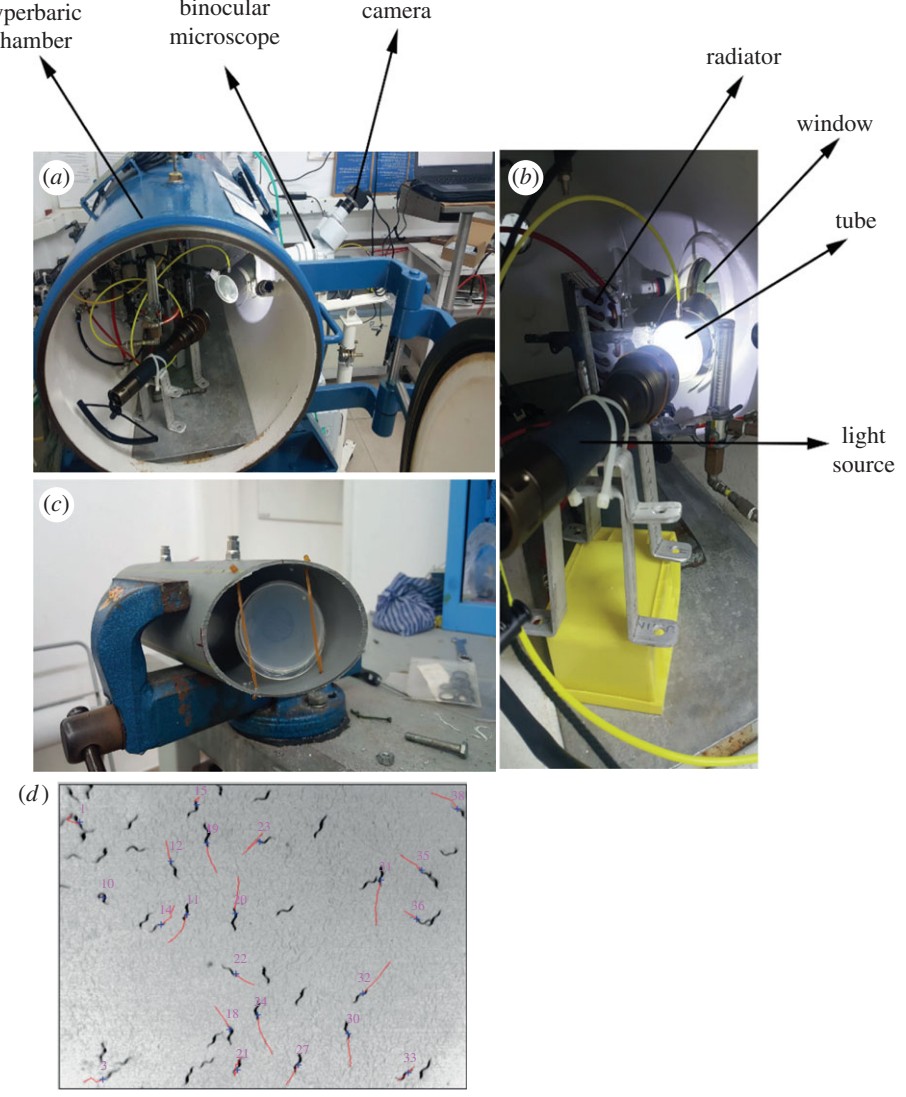

**Figure 1.** The hyperbaric chamber and the imaging tracking system for *C. elegans*. (*a*) The *C. elegans* imaging set-up contains the hyperbaric chamber and a microscope equipped with a camera. (*b*) The accessories installed inside the hyperbaric chamber include a radiator circulating cold water to keep the temperature stable during the experiment, a window for real-time imaging of the worms during the treatment, a tube to place the worm plate and a light source. (*c*) A zoom-in on the *C. elegans* agar plate inside the tube. (*d*) An example of worm tracks as extracted from the imaging movie using the multi-animal tracker system [28]. (Online version in colour.)

the pressure and that the second phase of the slowdown is moderate in 0.7 MPa when compared to 1.1 MPa. Upon decompression and recovery, the speed further decreased until reaching a steady low velocity. The animals were alive at the end of the recovery phase and after few hours returned to their normal speed. In the lowest tested pressure of 0.22 MPa (figure 2*a*, green line), we detected a small but non-significant increase in the average speed. We conclude that an increase in the velocity of WT *C. elegans* worms in the hyperbaric chamber was detected above 0.22 MPa and reached a maximum at 0.9 MPa.

We performed two control experiments in order to examine if temperature changes inside the chamber affect worm motility independent to compression. We first recorded the temperature inside the chamber along a 1.1 MPa cycle of a 'diving' protocol of WT animals in air. The temperature inside the chamber increased from 20°C to 22°C during compression and decreased to 16°C during decompression (electronic supplementary material, figure S1A). We next recorded worm motility along a temperature gradient (16–25°C) under normobaric conditions. The recorded

changes in speed were minor when compared to compression and decompression conditions (electronic supplementary material, figure S1B and table S2). These data validate that the pressure is indeed the main cause for the observed behavioural response and that changes in the temperature inside the chamber have minor effects.

### (b) Wild-type *Caenorhabditis elegans* are less responsive to a repeated cycle of compression–decompression–recovery

To examine if *C. elegans* worms adapt to the high pressure during repeated exposures, we performed two cycles of compression–decompression–recovery with 1 h compression in the maximal constant pressure of 1.1 MPa. In these experiments, we observed that the maximum normalized speed in the second cycle was significantly lower than the maximum speed in the first cycle ($3.1 \pm 0.7$ and $5.2 \pm 1.0$, respectively, $p < 0.0001$) and was measured at a lower pressure of 0.7 MPa compared to 0.9 MPa in the first cycle (figure 2*b* and electronic

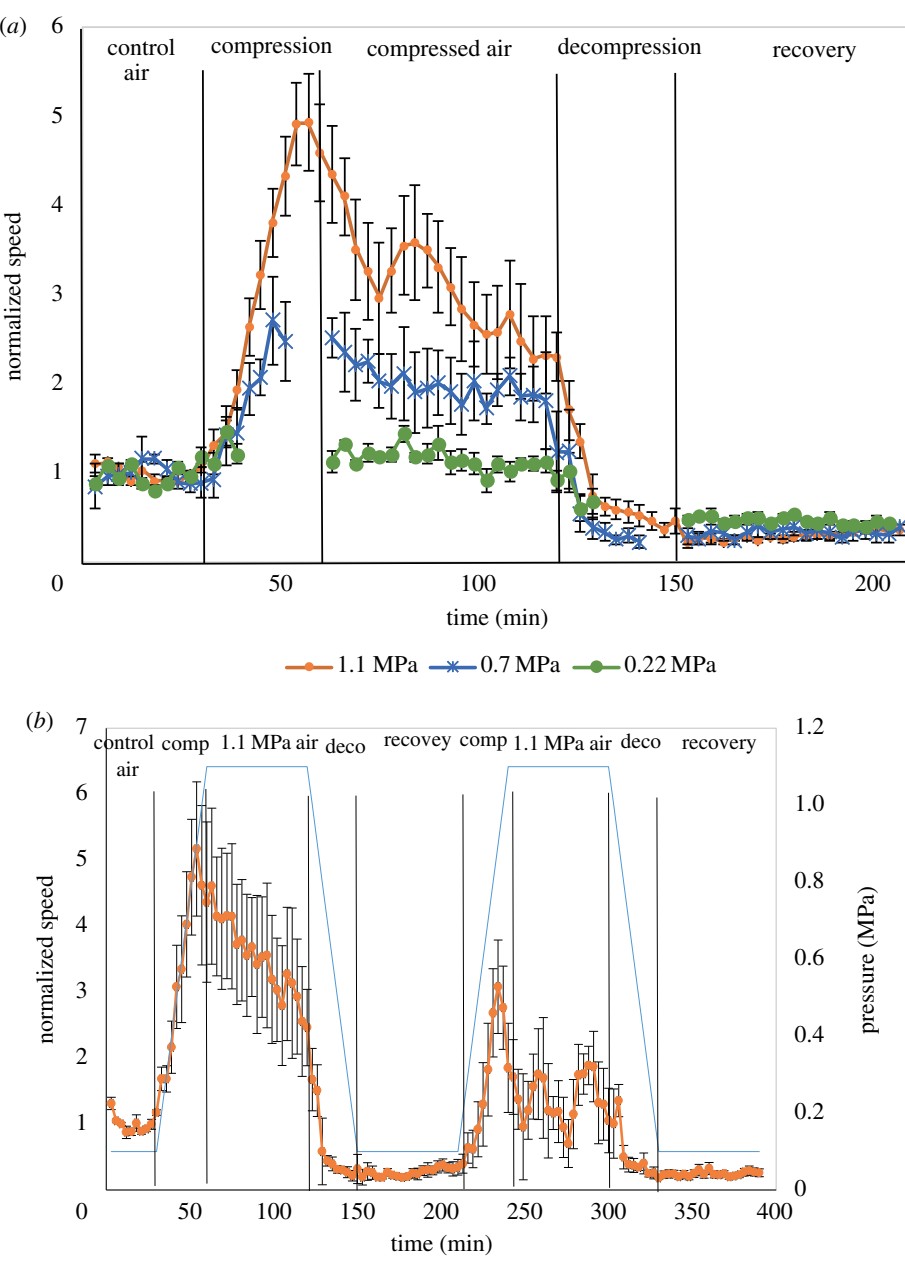

**Figure 2.** Biphasic behaviour of WT animals during compression in air. (*a*). WT *C. elegans* animals were imaged in the hyperbaric chamber before compression and during compression, decompression and recovery. Compression and decompression were performed in a rate of 0.03 MPa min$^{-1}$. The final pressures tested were 1.1 (orange), 0.7 (blue) and 0.22 (green) MPa. The *y*-axis is the average speed normalized to control normobaric speed (control air). The values represent mean (±s.e.) of 3–6 experiments with $n \geq 30$ tracks at each time point (electronic supplementary material, table S1). Statistical difference between the maximum speed in each of the three tested pressures and normobaric conditions was determined by two-sample *t*-test after first testing for an equal variance by *F*-test. The compression rate was identical and therefore the final pressure was achieved at an earlier time point in the cases of 0.7 and 0.2 MPa (shown as gaps). (*b*). WT animals were treated with two rounds of compression and decompression under final pressure of 1.1 MPa. The *y*-axis denotes the normalized speed which is the average speed normalized to control normobaric speed (control air). The *y*-axis pressure indicates the pressure in MPa at each time point (blue line). The values represent mean (±s.e.) of three experiments with $n \geq 30$ tracks at each time point (electronic supplementary material, table S3). Statistical difference between the maximal speed in the first and second round was determined by two-sample *t*-test after first testing for equal variance by *F*-test. (Online version in colour.)

supplementary material, table S3). In addition, the decrease in velocity was steeper in the second cycle and reached the baseline speed after 3 min in 1.1 MPa. This immediate decrease in velocity was in contrast to the gradual decrease in the first cycle. These data suggest that *C. elegans* can adapt to repeated pressure of 1.1 MPa following 1 h recovery between the first and the second cycle of compression. However, we cannot conclude with certainty that the animals were fully recovered following the 1 h break between the first and second cycle. If the animals were not fully recovered the immediate decrease

in speed during the second cycle could be because of the physiological conditions of the animals and not due to adaptation.

## (c) Dopamine mutants are less responsive to increased air pressure

The role of dopamine signalling in nitrogen narcosis has been demonstrated in rats [4]. To test if dopamine also plays a role in a simple model organism under pressure, we next examined the response to air pressure in two *C. elegans* mutants, both

defective in the dopamine pathway, *cat-2(e1112)* and *dat-1(ok157)*. CAT-2 encodes a tyrosine hydroxylase which catalyse the rate-limiting step in the synthesis of dopamine. The *cat-2(e1112)* strain is a dopamine-deficient mutant that exhibits a higher velocity in the presence of a food source in comparison to WT [18,22,30]. DAT-1 encodes a sodium-dependent dopamine transporter which terminate the action of dopamine by its high-affinity sodium-dependent re-uptake [31]. In the *dat-1(ok157)* mutant strain dopamine re-uptake is disrupted, but the animals exhibit locomotion rates similar to WT on standard substrate plates [27]. WT (figure 2), *cat-2(e1112)* and *dat-1(ok157)* animals were imaged before compression, during compression (1.1 and 0.22 MPa in air), at constant pressure, decompression and recovery. The average speed was normalized to the average control speed in air for each genetic background (figure 3a,b; electronic supplementary material, tables S4 and S5, movies S2 and S3). Both WT and the two dopamine mutants showed similar velocity patterns in 0.22 MPa with a moderate increase in normalized speed values upon pressure and decrease during decompression (figure 3a). Interestingly, *cat-2* mutants showed the highest response in this pressure (maximum normalized speed 2.4 ± 0.3, $p < 0.05$).

At 1.1 MPa, we observed dramatic differences between the dopamine mutants. The *dat-1(ok157)* mutant animals were responsive almost as WT and maintained the biphasic speed pattern including an increase during compression and a decrease during a steady pressure of 1.1 MPa. The maximum normalized speed of the *dat-1(ok157)* mutant animals at 1.1 MPa was significantly lower than the maximum speed of WT animals (3.8 ± 0.3 and 4.9 ± 0.5, respectively, $p < 0.0001$). In addition, the maximum speed in *dat-1(ok157)* was measured at a lower pressure of 0.7 MPa compared to 0.9 MPa in WT animals. This response resembled the adaptive WT response (figure 2b, second cycle). The *cat-2(e1112)* mutant animals showed markedly different behaviour. We observed a small increase in the normalized speed (1.7 ± 0.01) that was maintained constant during the entire 1 h compression at 1.1 MPa without a clear biphasic response (figure 3b). When the highest velocity values of WT and dopamine mutant animals under normobaric conditions and in 1.1 MPa air were compared without normalization to the basal speed of each genetic background, which is different between the strains, a significant increase in the maximum speed in the mutants was demonstrated under pressure (figure 3c). These results indicate that similarly to WT, the velocity of *cat-2* and *dat-1* mutants is increased under higher pressure. However, the biphasic response is impaired in *cat-2* mutants, suggesting that dopamine is required for the slowdown of the animals in response to continuous exposure to high-pressure levels.

### (d) Compressed oxygen induces minor changes in locomotion speed

The partial pressure of oxygen in 1.1 MPa compressed air is 0.22 MPa, compared to normobaric conditions in which oxygen pressure is approximately 0.02 MPa. To examine if the biphasic effect observed in 1.1 MPa air is due to the higher oxygen pressure that may result in oxygen toxicity, we analysed the speed under 0.22 MPa pure oxygen. In this condition *cat-2(e1112)* mutant animals increased their speed as soon as they were exposed to normobaric pure oxygen (control oxygen phase, figure 4a). In 0.22 MPa oxygen, we

detected a mild increase in the speed of *dat-1(ok157)*. During decompression, the speed decreased to values lower than control in the three strains, suggesting that decompression following compression in pure oxygen decreases worm motility (figure 4a; electronic supplementary material, table S6). We conclude that the biphasic velocity observed in compressed air in WT and *dat-1(ok157)* mutant animals is not a result of the higher partial pressure of oxygen in 1.1 MPa air.

### (e) Compressed nitrogen induces similar response as compressed air

To examine the contribution of nitrogen to the increased speed of the animals under pressure, we quantified animals' velocities under 1.1 MPa pressure of a mixture consisting of 95% nitrogen and 5% oxygen (figure 4b; electronic supplementary material, table S7). In this gas composition (partial pressure of nitrogen is 0.869 MPa) we observed similar velocity changes as compression of 1.1 MPa in air. The maximum increase in speed was observed at the maximum pressure in the three strains. When the highest velocity values of WT and dopamine mutant animals under conditions of 0.1 MPa in nitrogen and in 1.1 MPa nitrogen were compared without normalization to the basal speed of each genetic background, a significant increase in the maximum speed in the mutant strains was demonstrated under pressure (figure 4c), similarly to 1.1 MPa air (figure 3c). These data suggest that the response of *C. elegans* worms to pressure is dominated by the nitrogen component in the air.

## 3. Discussion

*Caenorhabditis elegans* has been a powerful model organism since the mid-1960s and repeatedly proven to be relevant to human physiological pathways and diseases since its genes and cells show extraordinary similarity to humans [32]. In this study, we examined *C. elegans* behaviour in deep-diving conditions, aiming to establish it as a multicellular model system that will enable the molecular mechanisms underlying nitrogen narcosis to be deciphered. Although nitrogen narcosis is a common phenomenon that every diver may develop during deep diving, the molecular mechanisms that mediate this high nitrogen toxicity and the genetic basis for the differences between individual divers are still unknown. We tested both WT and dopamine mutant strains under compressed air, oxygen and nitrogen. We found that WT animals respond to high pressure in air and nitrogen in a biphasic curve that was similarly reported for rats [29] and resembles human behaviour in nitrogen narcosis [4,33]. The animals responded to increased pressure by first increasing their locomotion speed. In the second phase of the response, their motility became slower. The slowdown started during compression and continued during incubation under the tested pressure, decompression and recovery phases. The increase in speed of WT *C. elegans* animals was detected at pressures above 0.22 MPa and reached a maximum speed at 0.9 MPa in air. In pressure above 0.9 MPa (maximum pressure tested was 1.1 MPa), the worms started to slow down. A similar effect, in which slowdown is observed under pressure and probably mimic nitrogen narcosis, was also measured in rats, albeit at a higher pressure of 3 MPa [29,34]. The lower pressure in which we observed

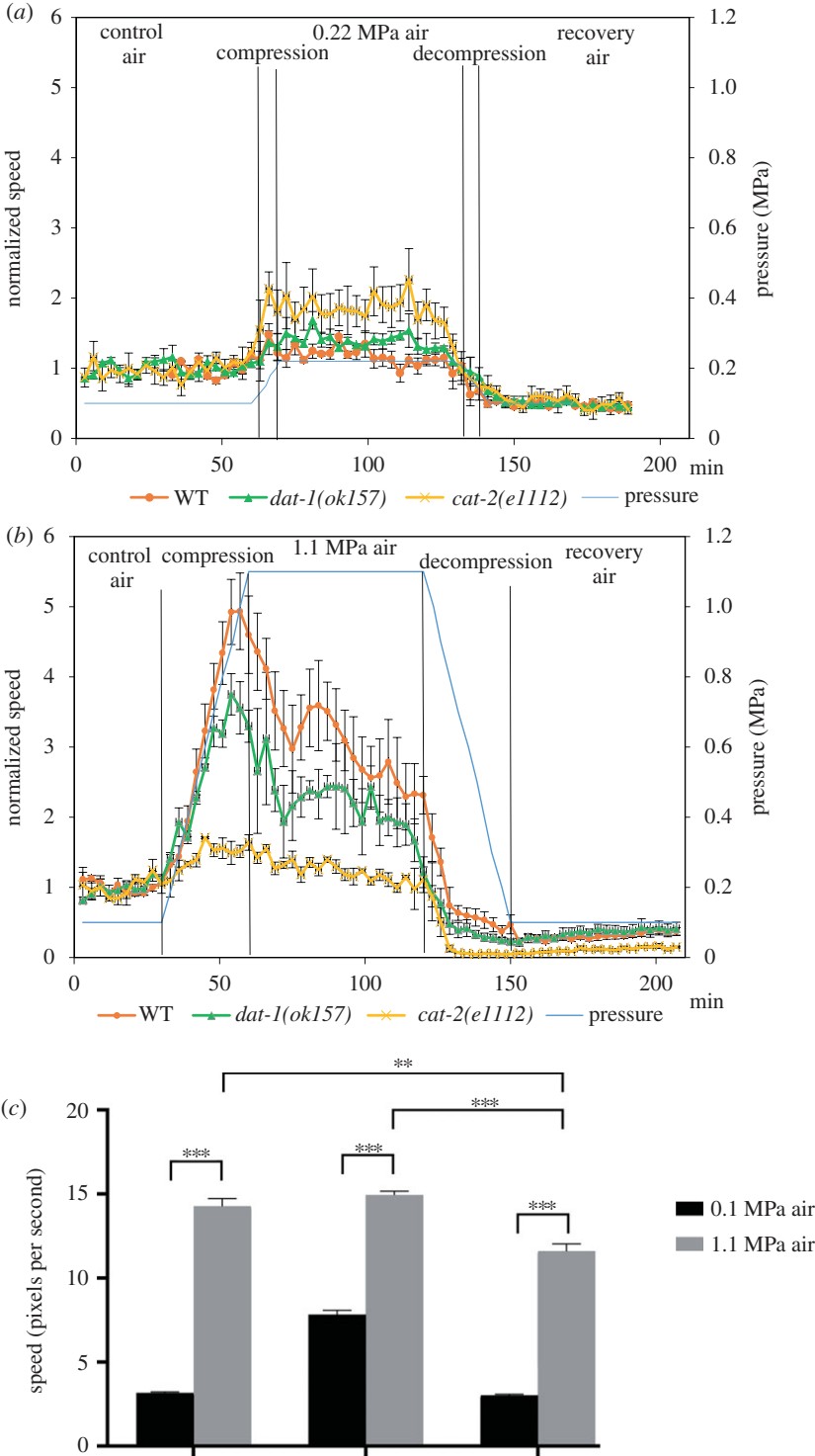

**Figure 3.** Dopamine is required for the biphasic response to compressed air. (*a,b*) WT, *cat-2(e1112)* and *dat-1(ok157)* worms were imaged in the hyperbaric chamber before compression and during compression, decompression and recovery. The final pressures tested were 0.22 MPa (*a*) and 1.1 MPa (*b*). The *y*-axis is the average speed normalized to the normobaric speed of each strain. The *y*-axis pressure indicates the pressure in MPa at each time point (blue line). The values represent mean (±s.e.) of 3–6 experiments with $n \geq 30$ at each time point (electronic supplementary material, tables S4 and S5). (*c*) Comparison between the highest speed values measured for WT and the dopamine mutants *cat-2(e1112)* and *dat-1(ok157)* under 1.1 MPa compressed air. The *y*-axis speed unit is pixels per second (1 pixel=16.67 μm). The values represent mean (±s.e.) of the highest speed recorded in the chamber at the indicated pressure in 3–6 experiments. Statistical difference between the maximum velocity in compressed air and normobaric conditions was determined by one-way ANOVA $F_{5,52} = 503.5$, *p*-value < 0.0001, followed by Tukey's multiple comparisons test $p < 0.0005$ (\*\*), and $p < 0.0001$ (\*\*\*). (Online version in colour.)

this response in worms when compared to rats can be explained by their smaller size and the lack of respiratory or circulatory systems, such that gases can diffuse rapidly into the cells. We suggest that this biphasic curve mimics nitrogen narcosis which begins with elevated excitement measured by increased motility in our system and ends in

loss of consciousness that we measured as a rapid decrease in locomotion speed. This biphasic curve also resembles the alcohol biphasic curve in humans where stimulant effects usually precede sedative outcomes [35]. Moreover, the examination of the response of WT worms to repeated exposures revealed that animals exposed to a second cycle of pressure

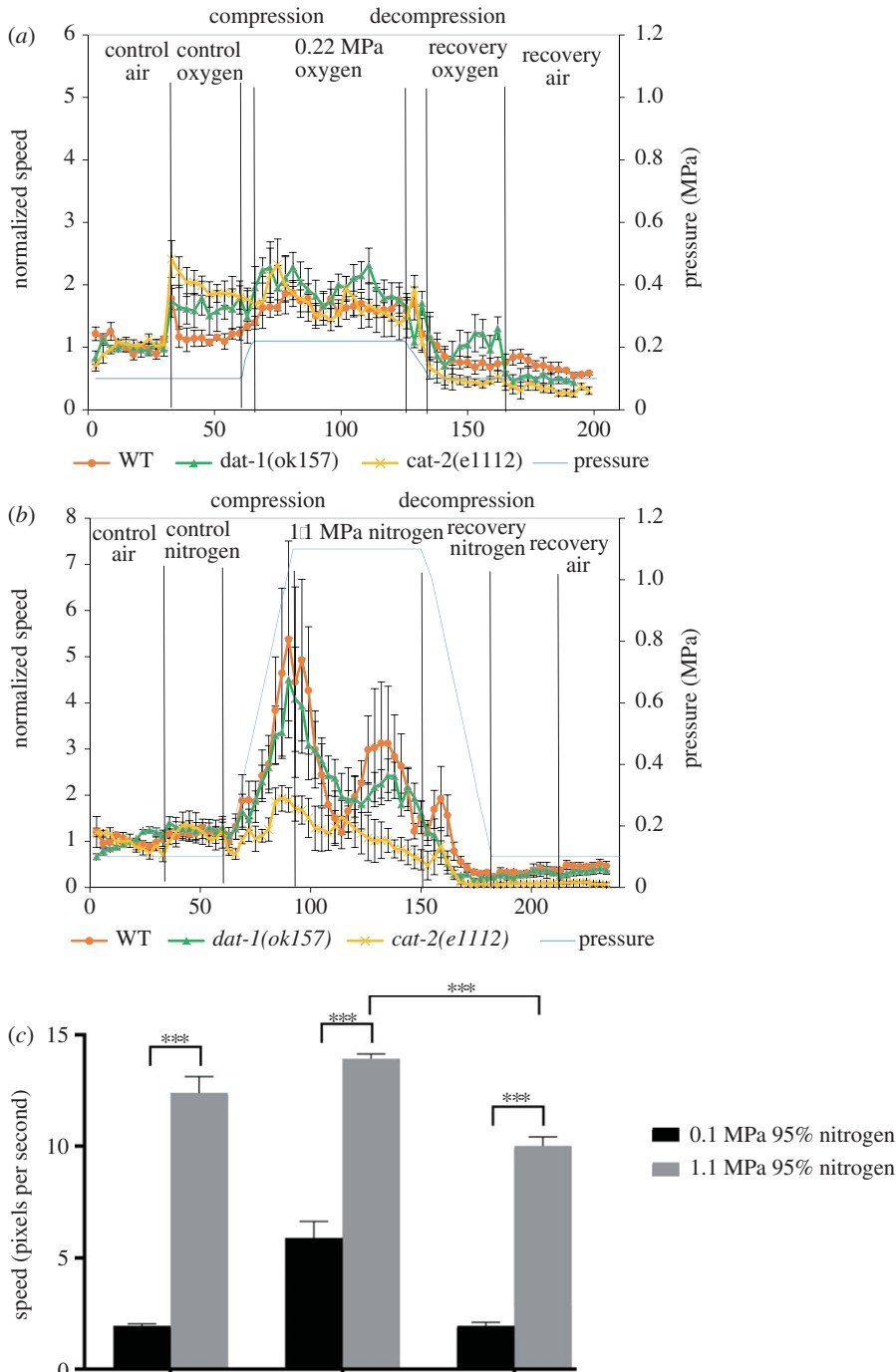

**Figure 4.** Compressed oxygen does not change *C. elegans* behaviour while compressed nitrogen induces the biphasic behaviour. (*a*) WT and mutant animals were imaged in the hyperbaric chamber before compression in air and in oxygen, and during compression, decompression and recovery in pure oxygen. The final pressure tested was 0.22 MPa oxygen. The *y*-axis is the average speed normalized to the normobaric speed of each strain. The *y*-axis pressure indicates the pressure in MPa at each time point (blue line). The values represent mean (±s.e.) of 3–6 experiments with $n \geq 30$ at each time point (electronic supplementary material, table S6). (*b*) WT and mutant animals were imaged in the hyperbaric chamber before compression and during compression, decompression and recovery in 95% nitrogen and 5% oxygen. The final pressure tested was 1.1 MPa nitrogen (95%). The *y*-axis is the average speed normalized to the normobaric speed of each strain. The *y*-axis pressure indicates the pressure in MPa at each time point (blue line). The values represent mean (±s.e.) of 3–6 experiments with $n \geq 30$ at each time point (electronic supplementary material, table S7). (*c*) Comparison between the highest speed values measured for WT and the dopamine mutants *cat-2(e1112)* and *dat-1(ok157)* under 1.1 MPa mixture of 95% nitrogen and 5% oxygen. The *y*-axis speed unit is pixels per second (1 pixel = 16.67 μm). The values represent mean (±s.e.) of the highest speed recorded in the chamber at the indicated pressure in 3–6 experiments. Statistical difference between the maximum velocity in compressed nitrogen and normobaric conditions was determined by one-way ANOVA $F_{5,87} = 139.5$, *p*-value < 0.0001, followed by Tukey's multiple comparisons test $p < 0.0001$ (***). (Online version in colour.)

show a weaker response suggesting an adaptive mechanism. This is similar to the observations made in rats and human divers, however, in our current experimental set-up, we could not determine if the animals were fully recovered

before the second exposure [8,9,34,36]. In rats, several studies had demonstrated that dopamine levels decrease following initial exposures, but these levels then increase following additional repetitive exposures to nitrogen narcosis

conditions [8]. However, motor deficiencies were not improved following the increase in dopamine levels, suggesting neurotoxicity or addiction effects. Furthermore, during recurrent exposures, GABA receptors undergo desensitization and glutamate levels that are coupled to increased NMDA receptor sensitivity decrease. In *C. elegans*, the different behaviours observed in the second exposure could be attributed to the increased levels of dopamine that led to faster entry into the slow motility phase. Using two dopamine mutant strains *cat-2(e1112)* and *dat-1(ok157)*, we revealed that dopamine is required for the biphasic response in compressed air or in compressed nitrogen. The response of *dat-1(ok157)* to 1.1 MPa resembles the response to the second exposure cycle in WT animals, suggesting that an increase in dopamine release (*dat-1* mutant animals accumulate dopamine) contributes to the adaptive response. As in other animals, dopamine signalling regulates *C. elegans* behaviour based upon previous experience [15]. Dopamine-deficient mutants alternate between low and high velocities more frequently and their ability to make small adjustments to speed is reduced compared to WT animals [17]. These observations support the broad distribution of average speeds we observed in the *cat-2(e1112)* mutant animals throughout the entire duration of the experiment.

To determine if the effects we measured are indeed dependent on nitrogen, we tested WT and dopamine mutant strains in 0.22 MPa of pure oxygen. There was only a mild change in the speed in response to compressed oxygen both in WT and dopamine mutant worms. Examination of worms in hyperbaric chamber in oxygen was previously performed in 100% $O_2$ at 40 psi (0.27 MPa) and showed that exposure to these conditions for 8 h increased the life expectancy of animals after returning to standard conditions [37]. On the other hand, when we examined the worms in 95% nitrogen and 5% oxygen in a pressure of 1.1 MPa, we observed a similar response to 1.1 MPa air. The partial pressure of nitrogen in the nitrogen-enriched mixture is 1.045 MPa while in the compressed air it is 0.869 MPa, suggesting that lower partial pressure of nitrogen may be sufficient to induce the *C. elegans* response.

In summary, in this study, we have demonstrated that high nitrogen pressure modulates *C. elegans* behaviour, a modulation that depends on dopamine. The significant differences observed between the highest speed values in pressure and basal conditions (figures 3*c* and 4*c*) indicate that dopamine is not the sole modulator of the worm behaviour under pressure, and it might be acting in parallel to other pathways. For example, hyperbaric nitrogen decreases the release of amino acids such as glutamate, glutamine and asparagine in the rat striatum [7]. In addition, the levels of the neurotransmitter serotonin increase in high nitrogen pressure [6]. The relation between the neurotransmitters and amino acids during nitrogen narcosis is still unknown.

Dopamine signalling in *C. elegans* is relevant to the understanding of human dopamine modulation due to the high conservation of proteins required for the synthesis, vesicular packaging, response and inactivation of dopamine [25]. Our data suggest that further studies of dopamine-dependent effects of nitrogen narcosis in the *C. elegans* model system might identify molecular and cellular mechanisms that are relevant to human divers, including the development of prediction tools for individual's risk assessment prior to diving at deep depths.

# 4. Methods

## (a) Strains

Strains were cultured and maintained at 20°C. The following strains were used in this study: N2 (Bristol, WT reference), CB1112: *cat-2(e1112)* II and RM2702: *dat-1(ok157)* III.

## (b) Exposure of *Caenorhabditis elegans* to hyperbaric conditions

Day-1 adult worms (approx. 50) were placed on a fresh NGM plate seeded with OP50 *Escherichia coli*. The plate was positioned inside a tube that was placed inside a 150 l hyperbaric chamber (Roberto Galeazzi, La Spezia, Italy) (figure 1). The tube was used to control the gas mixture the worms were exposed to, disregarding the compressed air used to elevate the ambient pressure in the chamber. The gas flow inside the tube was controlled by a needle valve located outside the chamber. An additional needle valve, also located outside the chamber, controlled the air flow to the entire chamber. The pressure inside the chamber was elevated at a slow rate equivalent to 0.03 MPa min$^{-1}$ to maintain a stable temperature during the experiment. In addition, a radiator with cold water (5°C) was placed in front of the air entrance to the chamber to cool down the incoming air. The gas in the chamber was either air, 100% oxygen or a mixture of 95% nitrogen and 5% oxygen. The experiment began by streaming air (1 l min$^{-1}$) to the back of the tube. Control movements were recorded for 30–60 min. After that, the chamber was compressed to the working pressure in a rate of 0.03 MPa min$^{-1}$. The working pressure was maintained for 1 h. Decompression to normobaric conditions was performed at a rate of 0.03 MPa min$^{-1}$ following by 30–60 min of recovery.

## (c) Temperature shift

The temperature in the chamber was elevated gradually from 16°C to 25°C (±0.2). The worms ($n > 150$) were kept under normobaric conditions. At each temperature, 5 min of habituation were followed by 15 min of imaging. The data were normalized to the average movement in 20°C.

## (d) Behavioural analysis

The animal movement was recorded during the treatment in the pressure chamber. The tracks of at least 30 worms were imaged for each time point by an MU900 (AmScope) camera, at a rate of 1 frame per second. From those frames, a video movie was made by FIJI ImageJ. Analysis of the movies was done using the 'MultiAnimalTrackerSuite' [28]. This program follows each track and cuts the data into blocks of 180 s. The video displays the total distance the worms made in intervals of 1 s. This time frame was found to be optimal to obtain reliable data through the natural movement changes.

Average speed units are pixels per second. Pixel size is 0.0167 mm.

## (e) Statistical analyses

Comparison between the speed of the worms under normobaric conditions and under each of the tested pressures and between the maximal speed in the first and second cycle of exposure were tested by two-sample *t*-test after first testing for equal variance by *F*-test (figure 2). Comparison between WT, *dat-1* and *cat-1* was done using one-way ANOVA (figures 3 and 4).

Data accessibility. It is provided as electronic supplementary material.

Authors' contributions. I.K. and B.A.: conception and design, acquisition of data, analysis and interpretation of data. E.I.: analysis of data, A.Z.: analysis of data, revising the manuscript critically. L.B.: conception and design, analysis and interpretation of data, writing the manuscript.

Competing interests. We declare we have no competing interests.

**Funding.** This research was supported by Israel Ministry of Defense (grant no. 4440838681) to L.B. and Israel Science Foundation ISF grant nos. 1878/15 and 2122/19 to L.B. and ISF grant no. 1300/17 to A.Z.

**Acknowledgement.** The *C. elegans* strains were provided by the Caenorhabditis Genetics Center (CGC), which is funded by NIH Office of Research Infrastructure Programs (P40 OD010440).

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
