## [Peer Review File · Proceedings of the Royal Society B: Biological Sciences]

Review History

RSPB-2020-2928.R0 (Original submission)

Review form: Reviewer 1

Recommendation

Major revision is needed (please make suggestions in comments)

Scientific importance: Is the manuscript an original and important contribution to its field?

Excellent

General interest: Is the paper of sufficient general interest?

Excellent

Quality of the paper: Is the overall quality of the paper suitable?

Acceptable

Is the length of the paper justified?

Yes

Should the paper be seen by a specialist statistical reviewer?

No

Do you have any concerns about statistical analyses in this paper? If so, please specify them explicitly in your report.

No

It is a condition of publication that authors make their supporting data, code and materials available - either as supplementary material or hosted in an external repository. Please rate, if applicable, the supporting data on the following criteria.

Is it accessible?

Yes

Is it clear?

Yes

Is it adequate?

Yes

Do you have any ethical concerns with this paper?

No

Comments to the Author

I reviewed the manuscript entitled: "Dopamine-dependent biphasic behavior under deep diving conditions in *C. elegans*" by Kirshenboim et al. This study presents the nematode *C. elegans* as a model system to study the molecular mechanisms that underly nitrogen toxicity in response to high pressure. The main finding of this paper is that wildtype *C. elegans* change their locomotion behavior in a biphasic mode in response to pressure of compressed air and nitrogen. Behavioral analysis of two strains with mutations in dopamine signaling suggests that the observed biphasic response is depended on dopamine. Data indicates that dopamine-deficient *cat-2* mutants lack the biphasic response to high pressure. Whereas, a mutation in the dopamine transporter encoding gene *dat-1*, mildly influenced this response likely due to elevated extrasynaptic dopamine. To determine whether the biphasic behavior is indeed dependent on nitrogen, the authors tested WT and dopamine mutants in pure oxygen and in nitrogen-oxygen mixture at pressures equal to that in compressed air. While no significant response was detected in compressed oxygen, animals responded to compressed nitrogen in a pattern similar to that observed in compressed air, suggesting that response observed under air pressure is due to nitrogen narcosis. Thus, this work allows the use of the powerful genetic toolkit of *C. elegans* to study the molecular basis of dopamine-dependent effects of nitrogen narcosis. The manuscript is well written, and experiments were carefully designed with proper controls; however, the following points should be taken into consideration before publication.

Major comments:

A) Phenotypic characterization:

The study is identifying an interesting phenotype in *C. elegans* in response to hyperbaric conditions termed "biphasic behavior"; however, essential information about the description of this behavior is missing in the results section. Please describe clearly the following: (1) what is the "biphasic behavior" and how it can be identified, characterized and calculated relative to pressure and time. (2) it is not obvious from Figure 2A that 0.7 MPa Pressure induces a biphasic behavior (as stated at line #146). This is probably due to the gap in 0.7 MPa measurements (the blue line in fig 2A) at the time (around 40min - 60min), which can also be seen for 0.2MPa (the green line in fig. 2A). It is not clear whether these gaps represent missing data?

In this context, the sentence in lines #151 & #152 is confusing, as it reads: ("Interestingly, when comparing the second phase of the biphasic response, the speed of the worms decreased during incubation at 1.1 MPa but remained relatively unchanged under constant pressure of 0.7 MPa."). If the speed of the worm was "relatively unchanged" as stated here, and a biphasic pattern is not obvious, does that means that there is no biphasic response at all at 0.7 MPa pressure point?

please elaborate.

B) Worm adaptation to high pressure during repeated exposures:

The authors concluded that the decline in speed in response to a second compression-decompression-recovery cycle was due to worms' adaptation to repeated pressure following 1hr recovery between the first and the second cycle of compression, as stated in line #177. However, there are two major concerns with this conclusion. First, it is highly possible that animals did not fully recover from the first pressure cycle and were still under stress. Thus, the decrease in animals' response in the second cycle might be due to decline in motor activity or residual effects from the first pressure cycle. According to the current experimental design, it appears that the recovery time of 1hr in this two-cycles experiment was too short. Indeed, animals would require more than one hour, probably few hours to return to their normal behavior after the first pressure cycle. The authors stated in lines #153 & #154: "The animals were alive at the end of the recovery phase and after few hours returned to their normal speed". Moreover, animals' velocity at the "recovery phase" is lower than their speed at "control air" phase in the experiment, indicating that they still under stress. I would suggest that in order to test whether *C. elegans* adopt to repeated pressure, animals must recover completely and return to their normal behavior before exposing them to a second pressure cycle. This would exclude confounding factors from the proceeding cycle. More extensive approach would require exposure to more than two cycles, with recovery intervals of sufficient time, to test whether the biphasic response would decline with repetitive exposure to pressure.

Minor points:

- In line #55, the sentence "accumulation of dopamine in the presynaptic cleft in dat-1" is probably not the best description of mutations in dat-1. Failure to reuptake dopamine in dat-1 mutant leads to dopamine accumulation in the extrasynaptic space (extracellularly); however, the term "presynaptic" here might imply the opposite. I would appreciate if the authors use a clearer sentence.
- o
- Figure 2A and 2B, please indicate whether normalization to basal speed was performed at the temperature gradient as in the pressure chamber?
- o
- The significant differences observed between highest speed values in pressure and basal conditions (Figures 3C & 4C) indicate that dopamine is not the sole modulator of the worm behavior under pressure, and it might be acting in parallel to other pathways. It would be nice if the authors can discuss this point briefly.
- Lines #200 & #201, The following sentence might be confusing: "maximal speed of WT animals (3.8 ± 0.3 and 4.9 ± 0.5 , respectively, $p < 0.0001$) and it was measured at a lower pressure of 0.7 MPa compared to 0.9 MPa in WT animals." What "it" refers too in this sentence? Do authors mean the "maximal speed"? please rephrase.
- lines #459 & #460, reference number #4 is not correctly listed:
"4. Neurochemistry of Pressure-Induced Nitrogen and Metabolically Inert Gas Narcosis in the Central Nervous System. In Comprehensive Physiology. Edited by:1579-1590."
- lines #474- #477: please remove the "]" at the end of this reference: "10. Gallo VP, Accordi F, Chimenti C, Civinini A, Crivellato E: Chapter Seven - Catecholaminergic System of Invertebrates: Comparative and Evolutionary Aspects in Comparison With the Octopaminergic System. In International Review of Cell and Molecular Biology. Edited by Jeon KW: Academic Press; 2016:363-394. vol 322.]"
- line #478: please correct the font in this reference to match the required journal style, and proper nomenclature of *Caenorhabditis elegans*: 11. Brenner S: THE GENETICS OF *CAENORHABDITIS ELEGANS*. Genetics 1974, 77:71-94.
- lines #479- #481, reference #12, please remove the unwanted characters .
"White JG, Southgate E, Thomson JN, Brenner S: The structure of the nervous system of the nematode *Caenorhabditis elegans*. Philosophical Transactions of the Royal Society of London. B, Biological Sciences 1986, 314:1-340."
- Please make sure that the terms "*Caenorhabditis elegans*" and "*C. elegans*" are in italic

throughout the references and the main text. Specifically, references numbers #13-30 where these terms show in the titles.

Review form: Reviewer 2

Recommendation

Accept with minor revision (please list in comments)

Scientific importance: Is the manuscript an original and important contribution to its field?

Good

General interest: Is the paper of sufficient general interest?

Acceptable

Quality of the paper: Is the overall quality of the paper suitable?

Good

Is the length of the paper justified?

Yes

Should the paper be seen by a specialist statistical reviewer?

No

Do you have any concerns about statistical analyses in this paper? If so, please specify them explicitly in your report.

No

It is a condition of publication that authors make their supporting data, code and materials available - either as supplementary material or hosted in an external repository. Please rate, if applicable, the supporting data on the following criteria.

Is it accessible?

Yes

Is it clear?

Yes

Is it adequate?

Yes

Do you have any ethical concerns with this paper?

No

Comments to the Author

The authors propose to use the *C. elegans* model system to unravel the molecular mechanisms responsible for decompression effect affecting divers by monitoring the speed of wild type animals and animals defective in dopamine synthesis and signaling on agar when exposed to varying nitrogen and oxygen pressures. When exposed to high nitrogen/air pressure, the speed of WT animals initially increases then gradually decreases. Once the pressure is relaxed, the animals' velocity decays to a very low level and never recovers its velocity under standard pressure. *dat-1* and *cat-2* mutants exhibited similar behavior like WT with smaller, yet significant, increases in velocity when pressurized. Under high nitrogen pressure, both WT and *cat-2* had similar speeds. It is hypothesized that nitrogen narcosis disrupts dopamine

pathways in the WT.

The paper is clearly written, and the reported observations are interesting. The paper's conclusion that the WT animals respond to elevated nitrogen pressure by increasing their speed is not fully supported since similar behavior (albeit to a lesser degree) is observed in the cat-2 mutants that are deficient in dopamine. Perhaps the authors could examine whether exogenous dopamine alters the behavior of cat-2.

Reports (e.g., PMID: 25902429) suggest that the mechanical properties of *C. elegans* are affected by external pressure. The authors may wish to examine the effect of pressure on the physiology of the worms and on the agar (?) substrate.

Minor comments

Line 141 and 362: Specify pressure in SI units – not m.

The authors should place the figures within the text and most importantly together with captions. Being unable to examine the figures concurrently with the captions makes the review process tedious.

Decision letter (RSPB-2020-2928.R0)

09-Jan-2021

Dear Dr Broday:

I am writing to inform you that your manuscript RSPB-2020-2928 entitled "Dopamine-dependent biphasic behavior under deep diving conditions in *C. elegans*" has, in its current form, been rejected for publication in Proceedings B.

This action has been taken on the advice of referees, who have recommended that substantial revisions are necessary. With this in mind we would be happy to consider a resubmission, provided the comments of the referees are fully addressed. However please note that this is not a provisional acceptance.

To upload a resubmitted manuscript, log into <http://mc.manuscriptcentral.com/prsb> and enter your Author Centre, where you will find your manuscript title listed under "Manuscripts with

Decisions." Under "Actions," click on "Create a Resubmission." Please be sure to indicate in your cover letter that it is a resubmission, and supply the previous reference number.

Sincerely,
Dr Sasha Dall
mailto: proceedingsb@royalsociety.org

Associate Editor
Board Member: 1
Comments to Author:

The paper focuses on the molecular mechanisms responsible for decompression effect affecting divers by studying *C. elegans* nematodes defective in dopamine synthesis and signaling under different levels of nitrogen and oxygen pressure. The paper was evaluated by two expert reviewers who made several important queries and suggestions that can improve the paper. I recommend the authors to respond to all of the points raised in detail. I think that this is a very interesting and promising approach and encourage the authors to resubmit their work.

Reviewer(s)' Comments to Author:

Referee: 1

Comments to the Author(s)

I reviewed the manuscript entitled: "Dopamine-dependent biphasic behavior under deep diving conditions in *C. elegans*" by Kirshenboim et al. This study presents the nematode *C. elegans* as a model system to study the molecular mechanisms that underly nitrogen toxicity in response to high pressure. The main finding of this paper is that wildtype *C. elegans* change their locomotion behavior in a biphasic mode in response to pressure of compressed air and nitrogen. Behavioral analysis of two strains with mutations in dopamine signaling suggests that the observed biphasic response is depended on dopamine. Data indicates that dopamine-deficient *cat-2* mutants lack the biphasic response to high pressure. Whereas, a mutation in the dopamine transporter encoding gene *dat-1*, mildly influenced this response likely due to elevated extrasynaptic dopamine. To determine whether the biphasic behavior is indeed dependent on nitrogen, the authors tested WT and dopamine mutants in pure oxygen and in nitrogen-oxygen mixture at pressures equal to that in compressed air. While no significant response was detected in compressed oxygen, animals responded to compressed nitrogen in a pattern similar to that observed in compressed air, suggesting that response observed under air pressure is due to nitrogen narcosis. Thus, this work allows the use of the powerful genetic toolkit of *C. elegans* to study the molecular basis of dopamine-dependent effects of nitrogen narcosis. The manuscript is well written, and experiments were carefully designed with prober controls; however, the following points should be taken into consideration before publication.

Major comments:

A) Phenotypic characterization:

The study is identifying an interesting phenotype in *C. elegans* in response to hyperbaric conditions termed "biphasic behavior"; however, essential information about the description of this behavior is missing in the results section. Please describe clearly the following: (1) what is the "biphasic behavior" and how it can be identified, characterized and calculated relative to pressure and time. (2) it is not obvious from Figure 2A that 0.7 MPa Pressure induces a biphasic behavior (as stated at line #146). This is probably due to the gap in 0.7 MPa measurements (the blue line in fig 2A) at the time (around 40min - 60min), which can also be seen for 0.2MPa (the green line in fig. 2A). It is not clear whether these gaps represent missing data?

In this context, the sentence in lines #151 & #152 is confusing, as it reads: ("Interestingly, when comparing the second phase of the biphasic response, the speed of the worms decreased during incubation at 1.1 MPa but remained relatively unchanged under constant pressure of 0.7 MPa."). If the speed of the worm was "relatively unchanged" as stated here, and a biphasic pattern is not

obvious, does that mean that there is no biphasic response at all at 0.7 MPa pressure point?
please elaborate.

B) Worm adaptation to high pressure during repeated exposures:

The authors concluded that the decline in speed in response to a second compression-decompression-recovery cycle was due to worms' adaptation to repeated pressure following 1hr recovery between the first and the second cycle of compression, as stated in line #177. However, there are two major concerns with this conclusion. First, it is highly possible that animals did not fully recover from the first pressure cycle and were still under stress. Thus, the decrease in animals' response in the second cycle might be due to decline in motor activity or residual effects from the first pressure cycle. According to the current experimental design, it appears that the recovery time of 1hr in this two-cycles experiment was too short. Indeed, animals would require more than one hour, probably few hours to return to their normal behavior after the first pressure cycle. The authors stated in lines #153 & #154: "The animals were alive at the end of the recovery phase and after few hours returned to their normal speed". Moreover, animals' velocity at the "recovery phase" is lower than their speed at "control air" phase in the experiment, indicating that they still under stress. I would suggest that in order to test whether *C. elegans* adopt to repeated pressure, animals must recover completely and return to their normal behavior before exposing them to a second pressure cycle. This would exclude confounding factors from the proceeding cycle. More extensive approach would require exposure to more than two cycles, with recovery intervals of sufficient time, to test whether the biphasic response would decline with repetitive exposure to pressure.

Minor points:

- In line #55, the sentence "accumulation of dopamine in the presynaptic cleft in *dat-1*" is probably not the best description of mutations in *dat-1*. Failure to reuptake dopamine in *dat-1* mutant leads to dopamine accumulation in the extrasynaptic space (extracellularly); however, the term "presynaptic" here might imply the opposite. I would appreciate if the authors use a clearer sentence.

o

- Figure 2A and 2B, please indicate whether normalization to basal speed was performed at the temperature gradient as in the pressure chamber?

o

- The significant differences observed between highest speed values in pressure and basal conditions (Figures 3C & 4C) indicate that dopamine is not the sole modulator of the worm behavior under pressure, and it might be acting in parallel to other pathways. It would be nice if the authors can discuss this point briefly.

- Lines #200 & #201, The following sentence might be confusing: "maximal speed of WT animals (3.8 ± 0.3 and 4.9 ± 0.5 , respectively, $p < 0.0001$) and it was measured at a lower pressure of 0.7 MPa compared to 0.9 MPa in WT animals." What "it" refers too in this sentence? Do authors mean the "maximal speed"? please rephrase.

- lines #459 & #460, reference number #4 is not correctly listed: "4. Neurochemistry of Pressure-Induced Nitrogen and Metabolically Inert Gas Narcosis in the Central Nervous System. In *Comprehensive Physiology*. Edited by:1579-1590."

- lines #474- #477: please remove the "]" at the end of this reference: "10. Gallo VP, Accordi F, Chimenti C, Civinini A, Crivellato E: Chapter Seven - Catecholaminergic System of Invertebrates: Comparative and Evolutionary Aspects in Comparison With the Octopaminergic System. In *International Review of Cell and Molecular Biology*. Edited by Jeon KW: Academic Press; 2016:363-394. vol 322.]"

- line #478: please correct the font in this reference to match the required journal style, and proper nomenclature of *Caenorhabditis elegans*: 11. Brenner S: THE GENETICS OF *CAENORHABDITIS ELEGANS*. *Genetics* 1974, 77:71-94.

- lines #479- #481, reference #12, please remove the unwanted characters .

"White JG, Southgate E, Thomson JN, Brenner S: The structure of the nervous system of the

nematode *Caenorhabditis elegans*. Philosophical Transactions of the Royal Society of London. B, Biological Sciences 1986, 314:1-340.”

- Please make sure that the terms “*Caenorhabditis elegans*” and “*C. elegans*” are in italic throughout the references and the main text. Specifically, references numbers #13-30 where these terms show in the titles.

Referee: 2

Comments to the Author(s)

The authors propose to use the *C. elegans* model system to unravel the molecular mechanisms responsible for decompression effect affecting divers by monitoring the speed of wild type animals and animals defective in dopamine synthesis and signaling on agar when exposed to varying nitrogen and oxygen pressures. When exposed to high nitrogen/air pressure, the speed of WT animals initially increases then gradually decreases. Once the pressure is relaxed, the animals' velocity decays to a very low level and never recovers its velocity under standard pressure. *dat-1* and *cat-2* mutants exhibited similar behavior like WT with smaller, yet significant, increases in velocity when pressurized. Under high nitrogen pressure, both WT and *cat-2* had similar speeds. It is hypothesized that nitrogen narcosis disrupts dopamine pathways in the WT.

The paper is clearly written, and the reported observations are interesting. The paper's conclusion that the WT animals respond to elevated nitrogen pressure by increasing their speed is not fully supported since similar behavior (albeit to a lesser degree) is observed in the *cat-2* mutants that are deficient in dopamine. Perhaps the authors could examine whether exogenous dopamine alters the behavior of *cat-2*.

Reports (e.g., PMID: 25902429) suggest that the mechanical properties of *C. elegans* are affected by external pressure. The authors may wish to examine the effect of pressure on the physiology of the worms and on the agar (?) substrate.

Minor comments

Line 141 and 362: Specify pressure in SI units – not m.

The authors should place the figures within the text and most importantly together with captions. Being unable to examine the figures concurrently with the captions makes the review process tedious.

Author's Response to Decision Letter for (RSPB-2020-2928.R0)

See Appendix A.

RSPB-2021-0128.R0

Review form: Reviewer 1

Recommendation

Major revision is needed (please make suggestions in comments)

Scientific importance: Is the manuscript an original and important contribution to its field?
Good

General interest: Is the paper of sufficient general interest?
Acceptable

Quality of the paper: Is the overall quality of the paper suitable?
Acceptable

Is the length of the paper justified?
Yes

Should the paper be seen by a specialist statistical reviewer?
No

Do you have any concerns about statistical analyses in this paper? If so, please specify them explicitly in your report.
No

It is a condition of publication that authors make their supporting data, code and materials available - either as supplementary material or hosted in an external repository. Please rate, if applicable, the supporting data on the following criteria.

Is it accessible?
Yes

Is it clear?
Yes

Is it adequate?
Yes

Do you have any ethical concerns with this paper?
No

Comments to the Author

I reviewed the manuscript RSPB-2021-0128 entitled "Dopamine-dependent biphasic behavior under deep diving conditions in *C. elegans*" as well as the authors' response to referees. The manuscript now clearly describes the interesting "biphasic behavior" in *C. elegans* in response to hyperbaric conditions. Furthermore, the authors addressed most of the concerns raised in the first review. I think that this manuscript has great merit. It will present the *C. elegans* as a tool to study the deep-diving conditions and nitrogen necrosis. More specifically, the presented data indicates *C. elegans* can be used to study the molecular basis of the biphasic response involving dopamine. However, I am afraid that the authors' conclusion that WT worms can adapt to a repeated pressure of 1.1 MPa within two pressure cycles is not strongly supported by data (Reviewer #1, comment #4). The manuscript still presents no evidence that animals can fully recover after "1 hr" between the first and second cycle of compression. In their response to this point, the authors referred to the table indicated in their reply in (comment #2 of reviewer #2); however, the data in that table was collected 4-6 hr after release from the hyperbaric chamber. Yet, it does not give us a clue about the behavioral or motor activity of the animals after a single hour of recovery inside the chamber.

As the authors indicated, technical difficulties might not allow for a well-designed experiment to test this hypothesis at this time. Thus, I would not recommend the inclusion of the data in lines # 225-235 and lines 357-362 (fig. 2), unless more evidence of a decline in response after full recovery and repeated pressure cycles are presented.

Decision letter (RSPB-2021-0128.R0)

29-Jan-2021

Dear Dr Broday:

Your manuscript has now been peer reviewed and the reviews have been assessed by an Associate Editor. The reviewers' comments (not including confidential comments to the Editor) and the comments from the Associate Editor are included at the end of this email for your reference. As you will see, the reviewers and the Editors have raised some concerns with your manuscript and we would like to invite you to revise your manuscript to address them.

Research ethics:

Use of animals and field studies:

It is a condition of publication that you make available the data and research materials supporting the results in the article (<https://royalsociety.org/journals/authors/author-guidelines/#data>). Datasets should be deposited in an appropriate publicly available repository and details of the associated accession number, link or DOI to the datasets must be included in the Data Accessibility section of the article (<https://royalsociety.org/journals/ethics->

policies/data-sharing-mining/). Reference(s) to datasets should also be included in the reference list of the article with DOIs (where available).

Please submit a copy of your revised paper within three weeks. If we do not hear from you within this time your manuscript will be rejected. If you are unable to meet this deadline please let us know as soon as possible, as we may be able to grant a short extension.

Best wishes,
Dr Sasha Dall
mailto: proceedingsb@royalsociety.org

Associate Editor Board Member

Comments to Author:

The authors provided a detailed response to reviewers' comments that is satisfactory for most part. I agree with Reviewer 1 who re-reviewed this version that the study presents interesting findings and introduces *C. elegans* as a useful model to study deep-diving conditions and nitrogen necrosis. However, there is one outstanding point as indicated in the letter by Reviewer 1. I suggest that the authors should either remove this part as suggested by the Reviewer or provide a clear statement that they cannot conclude with certainty that animals recovered fully after 1h, and explain how this affects the conclusions that can be made at this stage.

Reviewer(s)' Comments to Author:

Referee: 1

Comments to the Author(s).

I reviewed the manuscript RSPB-2021-0128 entitled "Dopamine-dependent biphasic behavior under deep diving conditions in *C. elegans*" as well as the authors' response to referees. The manuscript now clearly describes the interesting "biphasic behavior" in *C. elegans* in response to hyperbaric conditions. Furthermore, the authors addressed most of the concerns raised in the first review. I think that this manuscript has great merit. It will present the *C. elegans* as a tool to study the deep-diving conditions and nitrogen necrosis. More specifically, the presented data indicates *C. elegans* can be used to study the molecular basis of the biphasic response involving dopamine. However, I am afraid that the authors' conclusion that WT worms can adapt to a repeated pressure of 1.1 MPa within two pressure cycles is not strongly supported by data (Reviewer #1, comment #4). The manuscript still presents no evidence that animals can fully recover after "1 hr" between the first and second cycle of compression. In their response to this point, the authors referred to the table indicated in their reply in (comment #2 of reviewer #2); however, the data in that table was collected 4-6 hr after release from the hyperbaric chamber. Yet, it does not give us a clue about the behavioral or motor activity of the animals after a single hour of recovery inside the chamber.

As the authors indicated, technical difficulties might not allow for a well-designed experiment to test this hypothesis at this time. Thus, I would not recommend the inclusion of the data in lines # 225-235 and lines 357-362 (fig. 2), unless more evidence of a decline in response after full recovery and repeated pressure cycles are presented.

Author's Response to Decision Letter for (RSPB-2021-0128.R0)

See Appendix B.

Decision letter (RSPB-2021-0128.R1)

12-Feb-2021

Dear Dr Broday

I am pleased to inform you that your manuscript entitled "Dopamine-dependent biphasic behavior under deep diving conditions in *C. elegans*" has been accepted for publication in Proceedings B.

Open Access

Your article has been estimated as being 8 pages long. Our Production Office will be able to confirm the exact length at proof stage.

Paper charges

Sincerely,

Dr Sasha Dall

Associate Editor:

Board Member

Comments to Author:

I satisfied with the additional text in response to reviewer's comments. Congratulations on a very interesting study.

Appendix A

POINT BY POINT RESPONSE

Ms. ID RSPB-2020-2928

We thank the reviewers for their valuable comments and suggestions, which were most helpful in revising our manuscript. Below is a detailed point by point response to each of the comments.

Reviewer #1.

1. The study is identifying an interesting phenotype in *C. elegans* in response to hyperbaric conditions termed “biphasic behavior”; however, essential information about the description of this behavior is missing in the results section. Please describe clearly the following: (1) what is the “biphasic behavior” and how it can be identified, characterized and calculated relative to pressure and time.

We used the term “biphasic behavior” to describe two different behaviors of the worms that were observed under high pressure and depend on the time under this pressure. The initial response is an increase in speed and it is followed by a slowdown.

According to Balon et al (ref 30), the biphasic behavior is a behavior of hyperactivity (high locomotor and motor activity), and then a decrease of activity. The hyperactivity followed by decrease of activity is the biphasic behavior.

This phenomenon can be detected by tracking *C. elegans* behavior, as demonstrated in this study. By following the worms speed rate, we found a behavior as describe by Balon et al, a hyperactivity phase (increasing the speed rate during the compression) and then decrease in the speed rate (during exposure to high pressure, and/or even while compression is still in progress). In this case, the speed was calculated by tracking the amount of pixels the worm moved each second, and normalized by the average rate in normobaric conditions.

We now provide a detailed explanation of what we mean by “biphasic behavior” in page 5, lines 164 -167 in the revised version.

2. (2) it is not obvious from Figure 2A that 0.7 MPa Pressure induces a biphasic behavior (as stated at line #146).This is probably due to the gap in 0.7 MPa measurements (the blue line in fig 2A) at the time (around 40min – 60min), which can also be seen for 0.2MPa (the green line in fig. 2A). It is not clear whether these gaps represent missing data?

The gaps do not represent a missing data and appear because the final pressure was achieved at an earlier time point in the cases of 0.7 MPa and 0.2 MPa. In our experimental set-up the compression rate was identical in all the pressures tested. As the final exposure pressures vary, the graphical representation of synchronized exposure to pressure followed by decompression lead to gaps in the graphs. This explanation is now added in the figure legends (Figure 2A) of the revised manuscript in page 5.

3. In this context, the sentence in lines #151 & #152 is confusing, as it reads: (“Interestingly, when comparing the second phase of the biphasic response, the speed of the worms decreased during incubation at 1.1 MPa but remained relatively unchanged under constant pressure of 0.7 MPa.”). If the speed of the worm was “relatively unchanged” as stated here, and a biphasic pattern is not obvious, does that means that there is no biphasic response at all at 0.7 MPa pressure point? please elaborate.

According to our measurements, the speed of the worms at 0.7 MPa was first increased and then decreased and was stabilized following this decrease. This is the reason we still consider it as biphasic response. We now mention in page 6 lines 173-175: “These data show that the

level of the biphasic phase depends on the pressure and that the second phase of slowdown is moderate in 0.7 MPa when compared to 1.1 MPa”.

4. The authors concluded that the decline in speed in response to a second compression-decompression-recovery cycle was due to worms’ adaptation to repeated pressure following 1hr recovery between the first and the second cycle of compression, as stated in line #177. However, there are two major concerns with this conclusion. First, it is highly possible that animals did not fully recover from the first pressure cycle and were still under stress. Thus, the decrease in animals’ response in the second cycle might be due to decline in motor activity or residual effects from the first pressure cycle. According to the current experimental design, it appears that the recovery time of 1hr in this two-cycles experiment was too short. Indeed, animals would require more than one hour, probably few hours to return to their normal behavior after the first pressure cycle. The authors stated in lines #153 & #154: “The animals were alive at the end of the recovery phase and after few hours returned to their normal speed”. Moreover, animals’ velocity at the “recovery phase” is lower than their speed at “control air” phase in the experiment, indicating that they still under stress. I would suggest that in order to test whether *C. elegans* adopt to repeated pressure, animals must recover completely and return to their normal behavior before exposing them to a second pressure cycle. This would exclude confounding factors from the proceeding cycle. More extensive approach would require exposure to more than two cycles, with recovery intervals of sufficient time, to test whether the biphasic response would decline with repetitive exposure to pressure.

We thank the reviewer for this important comment. As this study was designed for a high temporal resolution analysis we imaged the worms at 1 frame/sec intervals along the entire experimental cycle. We expected minor additional changes till complete speed recovery. Moreover, we chose conditions that did not affect the long-term physiology of the worms (please see below our reply to comment #2 of reviewer #2). Still, we could not claim that the worms recovered completely after 1 hr. When we compared our protocol to protocols of repeated compression cycles in rats, the repeated compression was performed once in 24 hr (ref 9). Because *C. elegans* nematodes are much smaller in size compared to rats and their life span is much shorter, we estimated that 1 hr recovery is therefore relevant for a repeated cycle of compression. We agree that in future studies it is important to find specific markers to measure complete recovery and test repeated compression cycles following a complete recovery after each cycle. This is a long-term task which is critically important and is still an unknown factor in human recovery from nitrogen narcosis.

5. In line #55, the sentence “accumulation of dopamine in the presynaptic cleft in *dat-1*” is probably not the best description of mutations in *dat-1*. Failure to reuptake dopamine in *dat-1* mutant leads to dopamine accumulation in the extrasynaptic space (extracellularly); however, the term “presynaptic” here might imply the opposite. I would appreciate if the authors use a clearer sentence.

Thank you. Indeed, our description was not accurate.

1. In the abstract (line 55) we wrote: “extracellular accumulation of dopamine in *dat-1* mutant animals”.
2. In line 124 and line 244 we deleted “presynaptic”.

6. Figure 2A and 2B, please indicate whether normalization to basal speed was performed at the temperature gradient as in the pressure chamber?

This normalization was not performed based on the results presented in Figure S1B showing that temperature changes in the hyperbaric chamber induce a minor effect on locomotion speed.

7. The significant differences observed between highest speed values in pressure and basal conditions (Figures 3C & 4C) indicate that dopamine is not the sole modulator of the worm behavior under pressure, and it might be acting in parallel to other pathways. It would be nice if the authors can discuss this point briefly.

As described in the introduction, there are more predicted factors involved in the mechanism of nitrogen narcosis. For example, hyperbaric nitrogen decreases the release of amino acids such as glutamate, glutamine and asparagine in the rat striatum. The decrease in these amino acids may explain the decrease in dopamine levels, and the changes of locomotor and motor activity in nitrogen narcosis (ref 7). In addition, the levels of the neurotransmitter serotonin increase in high nitrogen pressure (ref 6). The relation between the neurotransmitters and amino acids during nitrogen narcosis is still unknown. We added this important point in the discussion section in page 12 lines 387-393.

8. Lines #200 & #201, The following sentence might be confusing: “maximal speed of WT animals (3.8 ± 0.3 and 4.9 ± 0.5 , respectively, $p<0.0001$) and it was measured at a lower pressure of 0.7 MPa compared to 0.9 MPa in WT animals.” What “it” refers too in this sentence? Do authors mean the “maximal speed”? please rephrase.

This sentence was now divided to two sentences in page 8 lines 255-258: “The maximum normalized speed of the *dat-1(ok157)* mutant animals at 1.1 MPa was significantly lower than the maximum speed of WT animals (3.8 ± 0.3 and 4.9 ± 0.5 , respectively, $p<0.0001$). In addition, the maximum speed in *dat-1(ok157)* was measured at a lower pressure of 0.7 MPa compared to 0.9 MPa in WT animals.”

9. lines #459 & #460, reference number #4 is not correctly listed: “4. Neurochemistry of Pressure-Induced Nitrogen and Metabolically Inert Gas Narcosis in the Central Nervous System. In Comprehensive Physiology. Edited by:1579-1590.”

This reference was corrected.

10. lines #474- #477: please remove the “]” at the end of this reference: “10. Gallo VP, Accordi F, Chimenti C, Civinini A, Crivellato E: Chapter Seven - Catecholaminergic System of Invertebrates: Comparative and Evolutionary Aspects in Comparison With the Octopaminergic System. In International Review of Cell and Molecular Biology. Edited by Jeon KW: Academic Press; 2016:363-394. vol 322.]”

This was corrected.

11. line #478: please correct the font in this refence to match the required journal style, and prober nomenclature of *Caenorhabditis elegans*: 11. Brenner S: THE GENETICS OF CAENORHABDITIS ELEGANS. Genetics 1974, 77:71-94.

This reference was corrected.

12. lines #479- #481, refence #12, please remove the unwanted characters . “White JG, Southgate E, Thomson JN, Brenner S: The structure of the nervous system of the nematode

Caenorhabditis elegans. Philosophical Transactions of the Royal Society of London. B, Biological Sciences 1986, 314:1-340.”

This reference was corrected.

13. Please make sure that the terms “*Caenorhabditis elegans*” and “*C. elegans*” are in italic throughout the references and the main text. Specifically, references numbers #13-30 where these terms show in the titles.

The references were corrected.

Reviewer #2.

1. The paper is clearly written, and the reported observations are interesting. The paper’s conclusion that the WT animals respond to elevated nitrogen pressure by increasing their speed is not fully supported since similar behavior (albeit to a lesser degree) is observed in the *cat-2* mutants that are deficient in dopamine. Perhaps the authors could examine whether exogenous dopamine alters the behavior of *cat-2*.

The *cat-2* mutants are well studied in *C. elegans* and indeed it has been previously shown that these animals are rescued by addition of dopamine to the media. For example in ref 17 (Omura et al., 2012) treatment with exogenous dopamine restored normal locomotion in *cat-2* mutant animals. We made efforts to measure the levels of dopamine in worms but this required decompression to release the plate from the chamber prior to lysis. In future studies we intend to generate fluorescent reporters for live imaging of the dopaminergic neurons inside the hyperbaric chamber along the complete compression cycle.

2. Reports (e.g., PMID: 25902429) suggest that the mechanical properties of *C. elegans* are affected by external pressure. The authors may wish to examine the effect of pressure on the physiology of the worms and on the agar (?) substrate.

We measured the effect of pressure on the physiology of the worms. We analyzed the animals using DIC microscopy 4-6 hr after release from the hyperbaric chamber. As summarized in the table below, abnormal physiological phenotypes as cell death in the anterior pharynx and egg laying defect were observed only after several hours under pressure. Only at 2 MPa we observed physiological damage after 1 hr pressure. In this study, when the pressure in the chamber reached the tested value, the worms were kept inside the chamber under this pressure for 1 hr. The data in the table demonstrate that the physiology of the worms was not affected in the conditions we used.

Pressure MPa (air)	Duration under pressure	Egg laying defect (Egl)	Cell death
0.5	5h	-	-
0.5	17.5 h	+	-
0.7	20 h	+	+
0.9	1 h	-	-
0.9	19 h	+	+
1.3	2 h	-	-
1.3	4 h	+	+
1.3	5h	+	+
1.3	19 h	+	+
2	1 h	+	+

Physiological effects of pressure on *C. elegans*. Egg laying defects and cell death in the anterior region of the pharynx following pressure treatment. At least 30 animals were tested in each condition.

3. Line 141 and 362: Specify pressure in SI units – not m.

We changed 3 meters/min into 0.03 MPa/min in lines 159, 160, 413,418, 420, 467 and 517.

4. The authors should place the figures within the text and most importantly together with captions. Being unable to examine the figures concurrently with the captions makes the review process tedious.

We now followed this helpful suggestion.

Appendix B

POINT BY POINT RESPONSE

MS Reference Number: RSPB-2021-0128

We thank the reviewer for his valuable comments and suggestions, which were most helpful in revising our manuscript. Below is a detailed point by point response to each of the comments.

Reviewer #1.

1. I reviewed the manuscript RSPB-2021-0128 entitled "Dopamine-dependent biphasic behavior under deep diving conditions in *C. elegans*" as well as the authors' response to referees. The manuscript now clearly describes the interesting "biphasic behavior" in *C. elegans* in response to hyperbaric conditions. Furthermore, the authors addressed most of the concerns raised in the first review. I think that this manuscript has great merit. It will present the *C. elegans* as a tool to study the deep-diving conditions and nitrogen necrosis. More specifically, the presented data indicates *C. elegans* can be used to study the molecular basis of the biphasic response involving dopamine.

We thank the reviewer.

2. However, I am afraid that the authors' conclusion that WT worms can adapt to a repeated pressure of 1.1 MPa within two pressure cycles is not strongly supported by data (Reviewer #1, comment #4). The manuscript still presents no evidence that animals can fully recover after "1 hr" between the first and second cycle of compression. In their response to this point, the authors referred to the table indicated in their reply in (comment #2 of reviewer #2); however, the data in that table was collected 4-6 hr after release from the hyperbaric chamber. Yet, it does not give us a clue about the behavioral or motor activity of the animals after a single hour of recovery inside the chamber. As the authors indicated, technical difficulties might not allow for a well-designed experiment to test this hypothesis at this time. Thus, I would not recommend the inclusion of the data in lines # 225-235 and lines 357-362 (fig. 2), unless more evidence of a decline in response after full recovery and repeated pressure cycles are presented.

As suggested by the editor we prefer to keep this data and add very clear statement that we have no evidence that the animals were fully recovered after 1 hr period between the two cycles. Our main claim in this section is that WT *C. elegans* are less responsive to a repeated cycle of compression-decompression-recovery and we only suggest that this could be adaptation. We think this data is still of interest to the reader even if the second response could reflect physiological stress and not adaptation. Accordingly, we added:

- 1) Lines 235-238: "we cannot conclude with certainty that the animals were fully recovered following the 1 hr break between the first and second cycle. If the animals were not fully recovered the immediate decrease in speed during the second cycle could be because of the physiological conditions of the animals and not due to adaptation".
- 2) Lines 364-365: "however in our current experimental setup we could not determine if the animals were fully recovered before the second exposure".